# Marketing Innovations in Industry 4.0 and Their Impacts on Current Enterprises

**Otakar Ungerman * and Jaroslava Dědková**

Technical University of Liberec Faculty of Economics, Department of Marketing and Trade Studentska 2, 46117 Liberec, Czech Republic

* Correspondence: otakar.ungerman@tul.cz; Tel.: +42-48535-2417

**Abstract:** This paper discussed the marketing innovations associated with Industry 4.0 and the effects that these innovative approaches cause. The main aim of the research was to discover the relationship between marketing innovations and their effects. Knowledge of this relationship can be used for the strategic planning of industrial companies in practice. The research methodology consisted of pilot research followed by primary research in industrial enterprises. The data were evaluated by descriptive statistics, statistical hypothesis, and correlation analysis. Through the research, the authors identified the importance of 17 innovative marketing tools and the strength of the use of 11 effects resulting from the implementation of these tools. The authors identified the relationships between tools and their implications in Industry 4.0 where a correlation was demonstrated. A list of 11 strategic objectives was created and, subsequently, a specific marketing mix proposal for each objective consisting of innovative marketing tools was as well. The results of this work enable enterprises involved in Industry 4.0 to better plan.

**Keywords:** Industry 4.0; marketing innovations; innovative marketing tools; impacts marketing innovations

## 1. Introduction

From the perspective of enterprises, innovations represent a key activity for their further development and increasing competitiveness within the current globalized market. When it comes to fundamental technological innovations that are linked to a sudden increase in labor productivity, this already concerns an industrial revolution. The first jump in labor productivity came in the 18th century in conjunction with the use of steam—the first industrial revolution. Subsequent labor productivity jumps were associated with the expansion of factory machine production—the beginning of the use of electricity and oil, automation, and subsequent digitization. Digitization is the main engine of the next expected jump in industry called Industry 4.0. According to the authors of References [1–3], Industry 4.0 involves complete digitization, robotization, and automation of most of today's human activities to ensure greater speed and efficiency of production, for more efficient use of materials, and for greener industry and human life. These are technological changes, such as the electronic transfer of information between machines (machine-to-machine), which solves problem situations without human involvement.

According to Reference [4], human capital is to be perfectly replaced in industry by autonomous robots and workers' professions will be abolished. The emergence of Industry 4.0 was first published in 2011 in Hanover, where in the publication "Industry Manifest 4.0", they presented the basic features of this industrial revolution [5,6]. Industry 4.0, however, is a broad term and is interpreted by various authors in different contexts. What the authors agree on is that Industry 4.0 will lead to fundamental changes in the economy, the working environment, and the development of skills [7]. At the time

of the establishment of Industry 4.0 to the present day, many authors (e.g., [8,9] ) have begun to call Industry 4.0 the Industrial Revolution 4.0.

The research results are significant for the whole European context, although data were collected only in the Czech Republic. The Czech economy is export based, where 80% of goods and services are exported [10]. At the same time Czech industry is strongly linked to multinationals property-wise. The share of foreign capital in Czech companies makes up half of the total capital. This interconnection of Czech companies with foreign corporations, e.g., Skoda auto versus Volkswagen leads to the conviction that the results can be generalized for the whole of developed Europe.

This paper deals with innovations, so it is necessary to define innovations correctly. According to the 2010 Eurostat update of the methodology used in the EU Business Innovation Activity Survey [11] (innovative enterprises are those which have introduced some of the innovations during the listed period:

(1) Product innovation—marketing of a new or substantially improved product or service;
(2) Process innovation—the introduction of a new or substantially improved method of production, provision of services, mode of supply, storage, distribution, introduction or substantial improvement of enterprise support activities;
(3) Marketing innovation—introduction of a new method of promotion, valuation or sale of products/services, significant changes in the aesthetic design or packaging of the offered products;
(4) Organizational innovation—introduction of a new way of organizing the supplier-customer relationship management, human resources or a new approach to the organization of external relations.

These four groups of innovations can be further divided into technological and non-technological. Technological innovations include product and process innovations and non-technological innovations include marketing and organizational innovations. This paper discusses marketing innovations that are categorized as non-technological innovations, nevertheless, it is impossible to ignore technology with them. There are big differences in the current characteristics of Industry 4.0-related marketing innovations and their impact. There are no current studies when it comes to examining the relationship between marketing innovation and impact. This situation led to the determination of the main research question:

Main research question:

How to use the current innovative marketing tools associated with Industry 4.0 for enterprise practice? This research question was divided into three research sub-questions:

1. How are innovative marketing tools used in Industry 4.0?
2. What are the implications of innovative marketing tools if enterprises implement them in their strategy?
3. What marketing mix does an enterprise involved in Industry 4.0 have to build to achieve a strategic goal?

Several hypotheses were defined in the research and were tested using statistical induction tests. The research questions were conditioned by performing the primary research, where the main objective was to find out what the implementation of Industry 4.0 brings to enterprises in the field of marketing innovations and what are the effects of these new trends. The methodology of the primary research is presented in Section 3.

## 2. Literature Review

After the research questions were set, a thorough investigation of scientific databases (Web of Science, Scopus, ProQuest) and other specialized literature was carried out. The research focused on three areas that were the subject of secondary research: innovation in Industry 4.0, innovation in marketing, and the effect that introducing these innovations brings to enterprises.

### 2.1. Innovation in Industry 4.0

As mentioned in the introduction, the EU classifies marketing innovation as a non-technological innovation group. However, if we want to define innovations related to Industry 4.0 that are based on technological progress, technology cannot be omitted from marketing innovations. Innovation in the era of industry 4.0 is perceived differently by authors. Lu [12] and Witkowski [13] emphasize that innovation is based on the development of the Internet and the existence of big data.

They state that the main innovations include the Internet of Things (IoT), cyber–physical systems (CPS), information and communication technologies (ICT), enterprise architecture (EA), and enterprise integration (EI), while the utilization of cybernetic systems is also reported by Marešová [14], who defines innovations in marketing as a flexible linking of products and services over the Internet or other network applications such as blockchain or a peer-to-peer network system.

The authors Zezulka and Veselý [15] divide innovation in Industry 4.0 into three groups that are interconnected and mutually influencing areas:

- Digitization and the integration of any production–business relationship: All links in the production chain will be able to access all necessary data. This can be very useful because, for example, machine builders, manufacturers of software and other production chain manufacturers and the entire production chain will be able to develop their products with the knowledge of the latest components that component manufacturers are yet to develop and test. The increase in digitization has an impact on companies' business activities, including their business models, in that they enable new forms of cooperation and lead to new products and services, as well as new forms of relationships with customers and employees. This digitization also places pressure on enterprises to consider their strategies and to systematically explore new business opportunities [16].
- Digitizing production and services: Based on data available through the cloud, manufacturers will be able to predict, for example, the failure or imminent failure of any manufacturer of electronic components that is needed for "their" production, machinery or equipment. The digitization of production data enables the optimization of demand, increases productivity, and allows efficient creation of values at the company's own production sites. The implementation of Industry 4.0 requires a high computing power to plan, process, simulate, and monitor production lines and to optimize and analyze data generated during the product lifecycle [17].
- New business models: They arise from the digitization and utilization of big data and lead to a precise definition and subsequent addressing of a homogeneous target group.

The authors of References [18–21] agree that Industry 4.0 works on six basic principles:

1. Interoperability: the ability of cyber–physical systems, people, and all smart factory components to communicate with each other through the Internet of Things and Services.
2. Virtualization: the ability to link physical systems to virtual models and simulation tools.
3. Decentralization: decision-making and control is carried out autonomously and in parallel in individual subsystems.
4. Ability to work in real-time: real-time compliance is a key requirement for any communication, decision-making, and control in real-world systems.
5. Service orientation: the preference of the computing philosophy of offering and using standard services, which leads to SOAs (service-oriented architectures).
6. Modularity and reconfigurability: Industry 4.0 systems should be maximally modular and capable of autonomous reconfiguration based on automatic situation detection.

An overview of the Industry 4.0 core tools has been compiled and published in the National Industry Initiative 4.0, where a team of authors with the support of the Ministry of Industry has compiled an overview of ten innovative tools of Industry 4.0.

1.  System integration—This is based on the interconnection of all links in the value chain from suppliers to the organizational structure of the manufacturing company itself to distribution to the end customer. The condition of the functionality of the interconnection is real-time data processing, information sharing, and continuous communication. At present, these connections are still inadequate and underdeveloped [22].

2.  Big Data Analysis—Big data is usually considered to be data in the range of peta bytes ($10^{15}$ bytes) or more that are currently at the edge of database technology capability. Examples are image data, text data from the Internet, business and security data, and combined multimodal data. Big data processing serves to optimize a company's production, related services, and distribution. The effort is to involve big data analysis for easier innovation, surpassing cheap mass production [23,24].

3.  Autonomous robots—robotic devices that work independently and do not require human control which are controlled by a program. Very often this type of robot works in collaboration with a person, where both actors complement each other. Nowadays, robots that are able to learn by themselves are beginning to gain ground and are developing the program themselves.

4.  Communication infrastructure—means using secure high-speed communication, primarily through wired and wireless networks. A link between products and networks is created where the necessary information is transferred among devices and machines throughout the production process.

5.  Data storage and cloud computing—These are server networks, each with a different function. Cloud solutions are used to store "big data", such as unstructured data. Clouds also help with planning new production. Using cloud solutions opens up opportunities for productivity growth and cost optimization. The big advantage is the possibility to share information among hundreds of branches of one company, e.g., about customers or sales structure [25].

6.  Additive production—it is the process of joining material according to 3D digital data, most often layer by layer. The product is produced quickly and precisely, even the most complex shapes such as printing a house. Additive technology makes it possible to produce diverse parts without the need for lengthy programming preparation [26]. Currently, 4D is being tested, which is a 3D product that can later change and reshape over time.

7.  Augmented Reality—connects the physical and virtual worlds. It extends the human perception of the world with new information that is not easily and quickly recognizable. Current applications are focused on smartphones and tablets that enable visualization of virtual tours, composing product groups, etc. Applications can be found in warehouse and logistics operations (barcode reading) in transport (traffic information) and in service (component visualization).

8.  Sensors—Sensors include methods and tools for measuring and sensing various variables that are important in an industrial automation system.

9.  Cybernetics and artificial intelligence provide key technologies for Industry 4.0 system solutions. These are the principles of organization, management, and decision-making as well as procedures to integrate autonomous systems.

10. New technologies—unused technologies will find their place in the Industry 4.0 process and new technologies will emerge, especially in the areas of biotechnology, information technology, and genetic technologies [27].

*2.2. Marketing Innovations*

Following the introduction of Industry 4.0 innovations, researchers focused their attention on industry-related marketing innovations. An exhaustive definition was used by Kotler [28] (p. 104):

> "By innovative (lateral) marketing we mean a sequence of work that, when applied to existing products, leads to the creation of new products or new services to meet new needs, bring new areas of use, new situations or discover new target groups of consumers. It is therefore

a process, offering a significant opportunity to create entirely new product categories or to form entirely new markets."

The same author equates innovative marketing to a process that requires a methodical approach. Its application to existing products or services brings possible innovations, for example, in the form of a new market or product category [28]. A concise definition of innovative marketing can also be found in the study entitled "Marketing Innovation: The Unheralded Innovation Vehicle to Sustained Competitive Advantage". The definition is as follows: "generation and implementation of new ideas for creating, communicating, and delivering value to customers and for managing customer relationships in ways that benefit the organization" [29] (p. 5). According to Reference [11], marketing innovation is the implementation of a new marketing method involving significant changes in the product or packaging design, product placement, product promotion or pricing. According to References [30,31], marketing innovation is defined as an innovation and a new method by which firms can sell to potential or existing customers. It includes significant changes in the implementation of various marketing strategies to increase marketing efficiency [32] allowing enterprises to gain a competitive edge and create value for shareholders.

The secondary research compiled an overview of some of the most important marketing trends of today, as perceived by world authors:

- Digital marketing—includes all marketing communications operating on the basis of digital technologies. Digital marketing trends include:

  ○ Artificial intelligence—autonomously evaluates the behavior of users on social networks. On a website, content automatically adapts to who's on the page. Artificial intelligence is used to write newsletters or posts on Facebook, increasing the clickthrough rate by tens of percent [33,34].

  ○ Conversational marketing—allows you to engage people in natural communication. Conversational will strengthen the brand and ultimately increase sales. This is a real-time conversation with a customer using chatbots. Chatbots find out everything a customer wants and prepare specific communication for them [35].

  ○ Personal brand/influencer—most commonly associated with video and youtubers. People do not want to follow enterprises but want to follow other people who are somehow interesting. This creates an "influencer." Today, an "influencer" is rewarded for their influence and product placement [36].

  ○ Search engine optimization (SEO)—social networks seemed to have killed SEO, but SEO is still very important and its importance has started to grow again. Link building that is used for SEO is focused on content quality and corporate blogs with videos and comments and client responses. In voice search, only the first position leads to a conversion. [37].

  ○ Omnipresence—Customers use multiple channels at once, and enterprises must spread their communications across all types of communication channels. This is a coordinated connection of all channels: email, YouTube, Instagram, Facebook, WhatsApp, and blogs with professional networks such as LinkedIn [38].

- Internet marketing, unlike digital marketing, always requires an internet connection. Ren, Xie, and Krabbendam [39] point out that internet marketing and marketing relationships have recently become the main focus of marketing innovations that companies use to achieve a sustainable competitive advantage. Similarly, Prahalad and Ramaswamy [40] and Son et al. [41] argue that the internet has changed the ways in which people live.

- Relationship marketing—focuses on long-term results to provide customers with long-term value and create high customer loyalty through building relationships at many levels, whether economic, social, technical or legal. It focuses on product benefits in a long-term horizon. It prefers intensive contact with customers with an emphasis on high responsibility towards them. [42].

- Mobile marketing—a form of advertising that is usually displayed on mobile devices, mostly phones or tablets. The huge increase in the popularity of smart phones with large displays has also caused the growing importance of mobile marketing. For mobile marketing, the abbreviation MAGIC can be applied: Mobile, Anytime, Globally, Integrated, Customized; that means using all the possibilities of digital marketing on a mobile device. It is therefore possible to influence the customer 24/7 [43].

These and other marketing innovations are used by enterprises to develop a strategy that meets a specific goal. The goal is to create new products and services that fill the gaps in the market and meet customers' unsuspecting needs. By implementing marketing innovation, a company can discover new product and service utilization, new market opportunities, and new groups of potential customers [25].

The authors Ohtonen [44] and Cummins et al. [45] add that the use of these marketing innovations can fulfill many diverse goals such as the introduction of new sales promotion methods, improvements in product packaging, innovation in promotion or the new use of media. It is evident that the scale of innovation activities in companies is determined by financial resources, the difficulty of introducing innovation to human resources, time and external factors such as political conditions, and the public perception of potential investments related to the introduction of innovation [46].

### 2.3. Impacts of Marketing Innovation

The impact of the onset of Industry 4.0 is already being reflected in enterprises. The main effect is the economic performance of enterprises, which can be measured in terms of accounting indicators such as cash flow and profitability. In addition, O'Sullivan and Abela [47] report that marketing performance is measured by return on assets (ROA) and return on investment (ROI). However, marketing performance can be measured by sales volume, revenue growth, and market share, while financial performance can be measured by profitability, revenue percentage, return on investment, profit, and profit growth. However, these are not only economic effects, but a whole range of effects associated with Industry 4.0. Industry 4.0 and marketing innovation provide a number of new opportunities and, if utilized, will have a strong impact on whole value chains. Among the key effects, according to References [7,48–51], include:

- Increase in labor productivity—the number of products/services and the production time are important to calculate labor productivity. Industry 4.0 leads to a dramatic increase in productivity, significantly reducing work time for the same production volume. Above all, there is a complete interconnection of the production process, including development and subsequent service. In factories, machines will be controlled by sensors, readers. and cameras. Robots automatically report maintenance to the maintenance staff. On the whole, the production process will be sped up and refined, while productivity will increase overall.
- The emergence of new business models—this is the basic principle of business, the way an enterprise creates and gains value from providing its services or selling products. New business models are linked to autonomous robotization in engineering. Industry 4.0 leads to new business models primarily related to direct selling [52]. New business models can resolve customer problems more effectively and find brand new customer segments [53]. Those business models, based on new technologies and big data, are focused on new services, value-linked ecosystems, and the approach to the customer to enable production to better respond to user-focused design and to better align with the processes and contexts involved in creating value for the customer [54].
- The cessation of "classic" jobs and the creation of "new" jobs—with the advent of Industry 4.0, people who devote themselves to automated operations will lose their jobs. It is not only the workers in assembly line production, but all who work in a routine way. Yet massive unemployment is not imminent. This is because a number of new positions will be created in services or industries where it is necessary to produce "customized goods".

- New workflows—process robotization, combining with IoT and other Industry 4.0 tools will lead to major workflow changes. This primarily involves the simplification of production. The change in workflows is related to changes in work organization. All this will translate into changes in the organizational structures of companies and the abandonment of the classic line models [55].

- New communication systems—these are models that are not linear but are circular, network models. With social networks, credit cards, and other Internet footprints, enterprises have perfect information about each individual. By processing big data, they can get to know their customer's behavior intimately and tailor their communication mix to fit their needs and, thus, achieve greater satisfaction. [56].

- Increased occupational safety—this impact is associated with the abolition of blue-collar jobs where there is the greatest risk of occupational injury. Leaving the risky work to robots will bring about a significant increase in occupational safety [57].

- Increasing competitiveness—the competitive strategy of companies involved in Industry 4.0 is primarily quality associated with precision processing. On the other hand, Industry 4.0 leads to higher production efficiency and reduced overall costs. The competitive advantage may then be the price [58].

- Increase PR—the involvement of companies in Industry 4.0 is used by enterprises in marketing. In their communication, they present the application of innovation, thereby building a better employer brand, which leads to better human capital. The presentation of the application of Industry 4.0 in the media leads to a better image in relation to the general public [12,47].

## 3. Materials and Methods

The primary methodology was designed to meet the research goal of "Finding out what Industry 4.0's implementation of marketing innovation brings to enterprises and the impact of these new trends". For a better overview of the procedure, both the conceptual research framework and the research evaluation methods are presented in Figure 1.

The basis of the research was a pilot project carried out in 2017. The aim of the pilot research was to identify innovative marketing tools and the effects they bring. For this identification, qualitative research inquiries with an in-depth view and a focus group were used. The results of the pilot research are presented in papers published in the Web of Science database [59,60].

In the Czech Republic where the research was conducted, there are a total of 31,966 enterprises registered in the Commercial Register [10], with 10 or more employees. Those enterprises comprised the core set (population). We contacted enterprises with more than 10 employees; smaller enterprises were excluded on the assumption that it is harder for them to implement innovations and those microenterprises would thus distort the sample set. The authors tried to obtain a representative sample of respondent and, thus, the minimal sample set size was calculated. The sample, with a ±5% sampling error at a 95% confidence interval, comprised 380 enterprises. During our research, we contacted a total of 605 enterprises; the return rate was 34%, i.e., 210 completed responses. Although the return rate of the responses was relatively high due to the fact of personal contact, the objective was not fulfilled. The sampling error was recalculated and, in conclusion, it may be said that 210 enterprises made up the representative sample with a 6.1% sampling error at a 95% confidence interval [60].

The structure of the enterprises by size can be broken down into enterprises with 10–49 employees, of which there are 21,101, making up 66% of enterprises. Enterprises with 50–249 employees, of which there were 8443, making up 26% of enterprises. Enterprises with 250 or more employees, of which there were 2422, making up 8% of enterprises [10]. The composition of the sample set was as follows: 10–49 employees 55%, 50–249 employees 30%, and 250 or more employees 15%. The distribution of respondents roughly corresponded to the composition of the core set, indicating that the sample was representative. The research included a sorting question, but the answers were not included in the assessment. The authors wanted to obtain a comprehensive view across all types of enterprises.

```
┌─────────────────────────────────────────────────────────────────────────┐
│                      Pilot research and evaluation:                       │
│       "Innovative Marketing in the Context of Industry 4.0" [59]          │
│   "The impact of marketing innovation on the competitiveness of           │
│   enterprises in the context of industry 4.0" [60].                       │
└─────────────────────────────────────────────────────────────────────────┘
```

| 1. RESEARCH QUESTION | 2. RESEARCH QUESTETION |
|---|---|
| **Marketing innovations in Industry** | **Impacts of marketing innovation** |
| Primary quantitative research: personal and electronic questioning | Primary quantitative research: personal and electronic questioning |
| Evaluation methods: -descriptive statistics -correlation analysis | Evaluation methods: -descriptive statistics -correlation analysis |

```
┌─────────────────────────────────────────────────────────────────────────┐
│                         3. RESEARCH QUESTION                              │
│               Relationship between IM tools and IM impacts                │
│                Evaluation methods: Correlation analysis                   │
└─────────────────────────────────────────────────────────────────────────┘
```

**Figure 1.** Conceptual framework of this study. Source: own.

The respondents were selected using systematic random sampling, which had to be performed in order to enable the use of statistical induction tests [61]. The parent university had access to the Bisnode database of companies, where enterprises are sorted alphabetically only, and which contains the entire population. The system used to select the enterprises involved contacting every 50th enterprise, which complied with the conditions of random research and enabled further statistical processing.

The pilot project was followed by primary research, which took place between June 2018 and March 2019. Two hundred and ten companies participated in the research, out of a total of 605 respondents, which was a return of 34%. The return on responses was relatively high, as the enterprises were approached in person and were subsequently sent a questionnaire electronically. It was necessary to obtain answers from key people, either owners, top management or the head of marketing in the enterprise. Enterprises with more than 10 employees were contacted and smaller enterprises were excluded on the assumption that innovation is more difficult to implement and that these micro-enterprises would distort the sample.

The data were collected using the IBM SPSS Data Collection software; IBM SPSS Advanced Statistics was used for the statistical assessment.

In the conceptual framework, it can be seen that research innovations in marketing and the effect they cause take place in parallel. The two investigations are then interconnected to establish the relationship. Methods of evaluating the detected data included:

(a) *Descriptive statistics* to determine and summarize information, process it in the form of graphs and tables, and calculate their numerical characteristics. Data processing methods used in the research included: frequency, arithmetic mean, standard deviation, median confidence interval, and minimum and maximum scale values.

(b) *Correlation analysis* depicts the statistical dependence of two quantitative variables. The aim of the correlation analysis was to determine the strength of the linear dependence between the frequency of use and the arithmetic mean of innovative marketing and the effects they caused. Correlation was also used to identify the dependence between marketing innovations and impact [62].

## 4. Final Evaluation of Research

Three research sub-questions were determined for this paper. According to these research questions, this section is divided into three subchapters, each giving an answer to one research question.

### 4.1. Innovative Marketing Tools

The first research question was set to identify the current use of innovative marketing in practice: "How are innovative marketing tools used during industry 4.0?" The tools examined were identified in the pilot research carried out by the qualitative method of data collection. In the pilot project, enterprises identified 17 tools that they perceived as part of Industry 4.0.

1.  Additive production—this is the formation of a physical product by gradual controlled addition of materials, such as metals, plastics, thermoplastics, glass. These include, in particular, casting and 3D printing [63]. According to Corsini, Aranda-Jan, and Moultrie [64] the standard metal and plastic machining industry cannot be replaced by additive manufacturing. Three-dimensional printing is not yet suitable for mass production and is only suitable for unique and complex products.
2.  Augmented reality—this is a representation of the real environment and the subsequent addition of visual information using 3D graphics. Thanks to 3D animation it is possible to present not only the appearance of the product, but also a demonstration of the product cut and its functionality [65].
3.  Virtual reality—allows the user to find himself in a simulated environment associated with user interaction. Virtual reality creates the illusion of a real world or a fictional world. In practice, it is used for the construction of buildings and cars, in the medical industry, or for computer games [66].
4.  Virtual currency (cryptocurrency)—this is based on the principle of peer-to-peer networks (client–client). This currency system has no superior control to regulate the currency. Virtual currency cannot be falsified due to the complex encryption. All transactions and accounts are public, which acts as protection and prevention of financial crime. The use of payment systems in practice is currently limited and used more as an investment [67].
5.  Autonomous distribution—in the consumer market, it is the delivery of the product directly to the customer's home, currently drones are the most used. In the industrial market, it is used in logistics, in transport among the intersections in the distribution channel. At the same time it is used in in-house logistics, where autonomous trucks provide production [68].
6.  Organizing events—this is a method of marketing communication connected with a form of performance, an experience that is associated with affecting emotions. In practice, events are divided into external communication, building relationships with stakeholders, and internal communication focused on their own employees. According to the authors Biswas and Suar [69] and Dabirian, Kietzmann, and Diba [70] event marketing is now at its peak again and its effectiveness is primarily in building employer branding.
7.  Relationship marketing—in contrast to transaction marketing, which is based on business needs, relationship marketing is based on customer needs. In practice, it is an effort to create, maintain, and expand strong and valuable relationships with stakeholders. Gillett [71] has proven that building customer loyalty and business success is strongly correlated.
8.  Product placement on shared multimedia—product placement is the placement of a product or brand in a movie, series, video, or photo to make it visible. The highest efficiency and effectiveness

according to Reference [65] is to use product placement on shared multimedia such as YouTube, Instagram or Vimeo.

9. Mobile app marketing—Kaplan [43] defines mobile marketing as a marketing activity conducted over the Internet to which consumers are constantly connected via a personal mobile device. Currently, companies see the greatest mobile marketing opportunities in applications, but this is no longer possible without app store optimization that will ensure a good position [72].

10. Quality function deployment—the aim is to incorporate the requirements of end customers into the final product of the company. The best way to apply QFD is to engage customers directly in product development. Lam and Bai [73] have proven that QFD leads to increased business competitiveness because the company develops and manufactures only products that the customer expects.

11. Product and packaging ecodesign—a systematic process of product design and development that, in addition to classic features such as functionality, places great emphasis on achieving a minimum negative impact of the product on the environment in terms of its entire life cycle. In practice, this means that the company will reuse all parts of the product at the end of its life cycle [74].

12. Internet of Things—this is so-called machine-to-machine communication. The product must have a built-in communication device to receive information from another device, process it, and provide it to another device. In practice, enterprises use IoT both in manufacturing, for example, in supplying production lines, or producing products that communicate with each other [75].

13. Circular economy—The circular economy separates economic growth from the need to extract new and rare materials. In reality, enterprises focus on material savings, recycling, reuse, and refurbishment. Lewandowski [76] and Velenturf and Purnell [77] have proven that business involvement in the circular economy is economically beneficial in the long run.

14. Guerilla and viral marketing—guerilla marketing is an unconventional form of marketing intended to shock. The goal is to get the maximum effect from minimal sources. Its low-cost use is primarily used by smaller enterprises. These aggressive attacks are mostly associated with a viral spread through social networks [78].

15. Advergaming—creating computer games for presenting enterprises or products. This is a link between the gaming business and marketing. These may be virtual worlds such as The Sims worlds or augmented games that combine reality with fiction such as Pokémon, where real "sponsored" sites serve as part of the game [79].

16. Employer branding—creating an employer's brand is a long-term and continuous process and consists in systematically creating and sharing positive employee experience. The main tool is sophisticated personnel communication with current, future, and former employees of the company. According to Reference [69], enterprises use this strategic tool to prevent employee turnover and attract the best possible future candidates.

17. Individual marketing using social media—also known as one-to-one marketing, this is a marketing strategy by which companies leverage data analysis and digital technology to deliver individualized messages and product offerings to current or prospective customers. According to Reference [80], enterprises need to use big data in conjunction with social media to carry out individual marketing.

These tools have been subjected to research. First, Table 1 presents descriptive statistics. The tools are listed in order of importance, from the most important to the least important.

**Table 1.** Evaluation of innovative marketing tools.

| Tool | $\bar{x}$ | n | SD | 95% Confidence Interval | | Median |
|---|---|---|---|---|---|---|
| | | | | Min. | Max. | |
| Tool 17 | 2.65 | 157 | 1.86 | 2.36 | 2.94 | 2 |
| Tool 7 | 2.91 | 137 | 1.772 | 2.61 | 3.21 | 2 |
| Tool 6 | 2.99 | 148 | 1.841 | 2.69 | 3.29 | 3 |
| Tool 13 | 3.26 | 154 | 1.885 | 2.96 | 3.56 | 3 |
| Tool 9 | 3.41 | 96 | 2.05 | 2.99 | 3.82 | 3 |
| Tool 8 | 3.44 | 116 | 1.971 | 3.08 | 3.8 | 3 |
| Tool 10 | 3.57 | 107 | 2.047 | 3.18 | 3.96 | 3 |
| Tool 14 | 3.61 | 137 | 1.884 | 3.29 | 3.93 | 3 |
| Tool 16 | 3.61 | 145 | 1.761 | 3.32 | 3.9 | 3 |
| Tool 2 | 3.89 | 102 | 2.009 | 3.5 | 4.29 | 3 |
| Tool 12 | 3.94 | 68 | 2.136 | 3.42 | 4.46 | 3 |
| Tool 3 | 4.01 | 79 | 2.047 | 3.55 | 4.47 | 4 |
| Tool 11 | 4.03 | 77 | 2 | 3.57 | 4.48 | 4 |
| Tool 1 | 4.15 | 78 | 2.064 | 3.69 | 4.62 | 4 |
| Tool 15 | 4.55 | 73 | 2 | 4.08 | 5.01 | 5 |
| Tool 4 | 5 | 54 | 2.249 | 4.39 | 5.61 | 6 |
| Tool 5 | 5.18 | 56 | 1.983 | 4.65 | 5.71 | 6 |

Source: own, ($n$ = 210, 1 = maximum importance, 7 = minimum importance).

This section may be divided by subheadings. It should provide a concise and precise description of the experimental results, their interpretation as well as the experimental conclusions that can be drawn.

If we divide the importance rating into two intervals (<1; 4) and (4; 7>), we get eleven tools that are important in the first interval and six that are not important. The second column shows the number of enterprises using each tool. The resulting values show that those tools that are perceived as important are used the most. The calculation of the correlation between the measure of importance and the frequency of use was −0.87234, which confirms the negative dependence.

The enterprises agree on the evaluation, which was confirmed by the standard deviation, where there were no large deviations. The identified tools can be combined into three groups. Each group contains tools that are identical in their use. These are:

(A) Tools targeting narrow homogeneous segments or individuals—enterprises identify innovative marketing as the most important tools that enable accurately targeting the most homogeneous segment. Even the most important was the tool that targets directly to individuals. Structured but also unstructured data are used for this direction, accessible from social networks. Industry 4.0 is about processing large amounts of unstructured information (big data). Focusing directly on the individual is associated with engaging customers directly in creating product design, organizing events, or reaching out with viral content.

(B) Promotion tools based on technological innovations—the second group of tools related to Industry 4.0 are technological marketing innovations, which include augmented reality, virtual reality, virtual currency, autonomous distribution, Internet of Things or advergaming. These tools are assessed by contradictory companies, which is evident from the higher dispersion. This may be due to the different levels of business involvement in Industry 4.0. The importance of these tools is clearly demonstrated despite lower assessments.

(C) Corporate social responsibility—The third group of marketing innovations are tools associated with corporate social responsibility. Although enterprises have classified them as Industry 4.0 related tools, they are non-technological tools. Their use is very common, and their importance is also relatively high. These are ecodesign, circular economy, and employer branding.

*4.2. Impacts of Innovative Marketing*

The second research question was "What impact do innovative marketing tools cause when enterprises implement them in their strategy?" The solution to this research question was a follow-up to the previous section. Identifying marketing innovations and their importance was a precondition for researching the impacts they cause. Impacts were also identified in the pilot project through qualitative research. Overall, 11 impacts were included in primary research.

The list of impacts was also determined in the pilot project, using qualitative research. Fifty enterprises participated in the pilot research, which, according to the available information, use innovative marketing in their business and present themselves as implementing Industry 4.0. The research was conducted in the form of personal questions, using the focus group method. The outputs were subjected to a content analysis, involving marketing experts and a practical expert specializing in the implementation of Industry 4.0 in enterprises. The responses to the open questions were first coded and then grouped into clusters. This procedure identified 11 main impacts of innovative marketing that the firms we questioned considered important. The resulting list may be described as containing fundamental impacts in the application of innovative marketing 4.0.

1.  Building PR and thus increasing the value of the enterprise. By implementing Industry 4.0 and innovative marketing in its program, the enterprise exhibits to the stakeholders and the surrounding area that it has a long-term vision. Implementation is usually linked to capital investment, which increases costs but also improves the image and value of the enterprise. This impact was most common in the replies half of all enterprises.

2.  Higher demands on employees.  Although marketing innovations are classified as non-technological, the implications of implementation are clearly linked to higher demands on employees. The Industry 4.0 philosophy will lead to a massive reduction in manual workers and a high demand for skilled people. Enterprises are already aware of the need to change the structure of their employees.

3.  Improving communication with customers. The identified marketing innovations lead to better knowledge of customers. With social networks, credit cards, and other Internet footprints, enterprises will have perfect information about each individual. By processing big data, they can get to know their customer's behavior intimately and tailor their communication mix to fit their needs and, thus, achieve greater satisfaction. Improved communication with the target group leads to the acquisition of new customers.

4.  Increasing the competitiveness of the enterprise. Respondents said that the implementation of innovative marketing and Industry 4.0 is in itself a competitive advantage which leads to assertion in a certain field compared to other enterprises. It is an increase in structural competitiveness resulting from ownership of assets or technology.

5.  Change in the amount of costs. Enterprises have agreed that the implementation of innovative marketing and Industry 4.0 is associated with cost changes. The replies showed that it was not possible to unequivocally claim that there was an increase in costs because some enterprises stated that there was a reduction in costs. The time factor plays a large role in costs. In the short term, due to the introduction of Industry 4.0, this is a cost which, in the long run, leads to cost reductions.

6.  Entering new markets. Thanks to an innovative approach, enterprises gain a competitive edge which aims to determine the growth strategy. These growth strategies often involve entering new markets. It is very often the internationalization of the enterprise, which concerns technologically developed countries. However, enterprises can expand their operations to other new segments where, for example, the expansion from the industrial to the consumer market has not yet functioned.

7.  Increasing labor productivity. Labor productivity is increased as a result of the introduction of improved technologies. Marketing innovations increase overall output, divided by work input.

Higher labor productivity leads to higher profits. This profit can then be used in the form of free capital for investments leading to further innovation. This cycle is driven by innovations associated with Industry 4.0.

8.  Change of distribution channels. It is primarily a systemic vertical integration that leads to property interconnection from production to sale. This situation leads to many acquisitions and mergers, which create large multinational corporations. Autonomous robotization will play a major role in distribution, especially in engineering. At the same time, Industry 4.0 will lead to the autonomous distribution of goods to the end customer, for example, using drones.

9.  Improving product quality. Better technology clearly implies an increase in product quality, for example, 3D printing while maintaining maximum accuracy in product manufacturing. Thanks to Industry 4.0, new materials are used with new properties that lead to improved product quality. The impact of implementation is clearly with products at a higher price level.

10. Changes in strategic planning. The implementation of Industry 4.0 leads to changes in long-term business planning. The basics of long-term planning is a strategic plan, where the vision of the enterprise is changed. The changes also affect other parts of the strategic plan, which are strategy and tactics. Enterprises see the great importance of digitization in the control of a strategic plan where, thanks to big data processing, the enterprise has a perfect overview of all outputs and hard data in context.

11. Change of company culture. Corporate culture can be characterized as a way of doing work and dealing with people. These are symbols of the enterprise (abbreviations, slang, dress code, symbols), hero promotion (serves as a model of ideal behavior), rituals (informal activities, formal meetings), and values that represent the deepest level of corporate culture. Respondents stated it was necessary to adapt to these changing needs of the market and clients as a result of Industry 4.0.

The identified impacts were subjected to research and subsequent statistical evaluation. Descriptive statistics were used once again, and the results are listed in Table 2 arranged in order of the strongest effects.

**Table 2.** The effects of innovative marketing on enterprises.

| Impact | $\bar{x}$ | $n$ | SD | 95% IS | | Median |
|---|---|---|---|---|---|---|
| | | | | Min. | Max. | |
| Impact 6 | 2.58 | 192 | 1.59 | 2.35 | 2.8 | 2 |
| Impact 4 | 2.74 | 191 | 1.675 | 2.5 | 2.98 | 2 |
| Impact 1 | 2.78 | 175 | 1.58 | 2.54 | 3.01 | 2 |
| Impact 9 | 3.07 | 171 | 1.696 | 2.81 | 3.33 | 3 |
| Impact 5 | 3.13 | 183 | 1.664 | 2.88 | 3.37 | 3 |
| Impact 7 | 3.26 | 160 | 1.982 | 2.95 | 3.57 | 3 |
| Impact 2 | 3.39 | 163 | 1.783 | 3.12 | 3.67 | 3 |
| Impact 8 | 3.43 | 172 | 1.648 | 3.18 | 3.68 | 3 |
| Impact 10 | 3.52 | 174 | 1.733 | 3.26 | 3.78 | 3 |
| Impact 11 | 3.83 | 156 | 1.911 | 3.52 | 4.13 | 3 |
| Impact 3 | 3.86 | 170 | 1.718 | 3.6 | 4.12 | 4 |

Source: own, 1 = strongest impacts; 7 = weakest impacts.

If the effects of innovative marketing are again broken down by a threshold of 4, it can be stated that all of the effects were identified by enterprises as very strong when they were above the threshold. At the same time, the strength of the impacts correlated significantly with the number of enterprises. The correlation coefficient between the impact strength and the number of enterprises that were identified as having impact effects was −0.7990. At the same time, there was a broad consensus on the assessment, as evidenced by the low standard deviation for all impacts.

### 4.3. The Relationship of Innovative Marketing and Its Impact

After identifying current innovative marketing tools and their impacts, the authors focused on the relationship between the resulting innovative tools and impacts. The basic idea was that the identified effects from Section 4.2 can be viewed in reverse. Effects arising from the implementation of marketing innovations can be understood as the strategic goals of the enterprise. In practice, instead of impacts, an enterprise can include the identified impacts in its plan as possible targets. For example, the identified impact of "Building Public Relations and Branding" can be included by an enterprise in its strategic plan as a long-term goal.

If all the identified impacts change to goals, we can meet each goal with the innovative marketing tools identified. For each goal, it is possible to build an individual marketing mix, by calculating the correlation between tools and impacts. The correlation data were collected by a systematic random selection of respondents, as described in the methodology. The research question "What marketing mix does an enterprise involved in Industry 4.0 have to build to achieve a certain strategic goal" was identified.

To determine the correlation between tools and impacts, pairs of scaled vectors were first created. Pairs where respondents said they did not use the tool or impact were omitted. The created pairs were subjected to correlation analysis, where the influence of the evaluation of the importance of individual tools of innovative marketing on the evaluation of individual impacts was examined. In this way, $17 \times 11$ pairs of vectors were generated, and their mutual state was analyzed. The following model was investigated:

H0: $r_{Toolx, Impactx} = 0$

HA: $r_{Toolx, Impactx} \neq 0$

Table 3 summarizes the resulting p-values for each correlation test. The reliability coefficient is set to $\alpha = 0.05$, where the *p*-value is 0.05, a statistically significant relationship is indicated.

**Table 3.** Resulting *p*-values of the correlate analysis test of the pair tool $\times$ impact.

| IM Tools | Impact | | | | | | | | | | | |
|---|---|---|---|---|---|---|---|---|---|---|---|---|
| | 1 | 2 | 3 | 4 | 5 | 6 | 7 | 8 | 9 | 10 | 11 | Σ |
| 1 | 0.813 | 0.728 | 0.917 | 0.997 | 0.275 | 0.493 | 0.044 | 0.637 | 0.036 | 0.615 | 0.076 | 2 |
| 2 | 0.052 | 0.555 | 0.068 | 0.362 | 0.005 | 0.433 | 0.038 | 0.014 | 0.141 | 0.114 | 0.007 | 4 |
| 3 | 0.675 | 0.855 | 0.092 | 0.231 | 0.083 | 0.260 | 0.004 | 0.084 | 0.018 | 0.640 | 0.133 | 2 |
| 4 | 0.472 | 0.006 | 0.289 | 0.598 | 0.101 | 0.364 | 0.044 | 0.774 | 0.048 | 0.891 | 0.803 | 3 |
| 5 | 0.237 | 0.037 | 0.152 | 0.599 | 0.474 | 0.352 | 0.654 | 0.636 | 0.338 | 0.755 | 0.253 | 1 |
| 6 | 0.021 | 0.745 | 0.225 | 0.000 | 0.153 | 0.007 | 0.811 | 0.003 | 0.130 | 0.039 | 0.232 | 5 |
| 7 | 0.090 | 0.369 | 0.005 | 0.169 | 0.136 | 0.068 | 0.496 | 0.037 | 0.033 | 0.040 | 0.015 | 5 |
| 8 | 0.568 | 0.432 | 0.328 | 0.555 | 0.411 | 0.022 | 0.335 | 0.112 | 0.201 | 0.076 | 0.379 | 1 |
| 9 | 0.624 | 0.563 | 0.049 | 0.509 | 0.639 | 0.010 | 0.998 | 0.022 | 0.053 | 0.333 | 0.816 | 3 |
| 10 | 0.108 | 0.915 | 0.371 | 0.544 | 0.781 | 0.724 | 0.038 | 0.399 | 0.472 | 0.806 | 0.543 | 1 |
| 11 | 0.023 | 0.599 | 0.322 | 0.064 | 0.035 | 0.744 | 0.465 | 0.846 | 0.967 | 0.887 | 0.047 | 3 |
| 12 | 0.508 | 0.736 | 0.144 | 0.570 | 0.011 | 0.057 | 0.206 | 0.167 | 0.075 | 0.255 | 0.225 | 1 |
| 13 | 0.233 | 0.925 | 0.458 | 0.063 | 0.009 | 0.006 | 0.103 | 0.082 | 0.114 | 0.112 | 0.013 | 3 |
| 14 | 0.036 | 0.598 | 0.056 | 0.008 | 0.000 | 0.000 | 0.873 | 0.051 | 0.003 | 0.002 | 0.077 | 6 |
| 15 | 0.013 | 0.381 | 0.021 | 0.262 | 0.590 | 0.251 | 0.238 | 0.536 | 0.473 | 0.805 | 0.564 | 2 |
| 16 | 0.060 | 0.043 | 0.112 | 0.222 | 0.049 | 0.051 | 0.017 | 0.001 | 0.436 | 0.017 | 0.177 | 5 |
| 17 | 0.000 | 0.613 | 0.031 | 0.002 | 0.399 | 0.000 | 0.050 | 0.034 | 0.068 | 0.101 | 0.131 | 6 |
| Σ | 5 | 3 | 4 | 3 | 6 | 6 | 7 | 6 | 5 | 4 | 4 | |

Source: own. (IM = Innovative Marketing).

The resulting table shows that each impact was related to several tools. There were a minimum of three and a maximum of seven tools. The correlation analysis confirmed that all tools were associated with certain impacts. Overall, the two tools "guerilla and viral marketing" and "individual marketing using social media" correlated most with the six impacts. By contrast, four innovative tools correlated with only one impact.

If an enterprise had one of the identified impacts among its strategic objectives, it can successfully use the identified tools. These are the tools that companies had identified as an effective strategy leading to the set goals. If innovative marketing tools are divided into the three groups (A–C) when identifying innovative marketing tools, this breakdown does not apply when compiling individual marketing mixes, as shown in Table 4.

**Table 4.** Strategic objective and marketing mix. (SM: social media).

| Strategic Objective | Marketing Mix of Innovative Tools | | |
| --- | --- | --- | --- |
| | A. Tools Targeting Narrow Homogeneous Segments or Individuals | B. Promotion Tools Based on Technological Innovations | C. Corporate Social Responsibility |
| 1. Building public relations and brand | Organizing events, guerilla and viral marketing, Individual marketing using SM | Advergaming | Ecodesign of product and packaging |
| 2. Higher demands on employees | | Virtual currency Autonomous distribution | Employer branding |
| 3. Improving communication with customers | Relationship marketing Individual marketing using SM | Advergaming Mobile app marketing | |
| 4. Increasing competitiveness | Organizing events, Guerilla and viral marketing, Individual marketing using SM | | |
| 5. Change in the amount of costs | Guerilla and viral marketing | Augmented reality Internet of Things | Circular economy Ecodesign of product and packaging |
| 6. Entering new markets | Organizing events Product placement on shared multimedia Guerilla and viral marketing Individual marketing using SM | Mobile app marketing | Circular economy Ecodesign of product and packaging |
| 7. Increasing labor productivity | Quality function deployment Individual marketing using SM | Virtual reality Additive production Virtual currency Augmented reality | Employer branding |
| 8. Change of distribution channels | Organizing events Relationship marketing Individual marketing using SM | Mobile app marketing Augmented reality | Employer branding |
| 9. Product quality improvement | Relationship marketing Guerilla and viral marketing | Virtual currency Virtual reality Additive production | |
| 10. Changes in strategic planning | Organizing events Relationship marketing Guerilla and viral marketing | | Employer branding |
| 11. Change of company culture | Relationship marketing | Augmented reality | Circular economy Ecodesign of product and packaging |

The results of the research have shown that it is not only tools from one group that can be used to implement the strategy, but a toolkit composed of all three groups. In conclusion, there is a clear link between the marketing tools associated with Industry 4.0 and the impacts associated with this time of major change. These relationships can be used by enterprises for accurate and successful strategy development.

## 5. Discussion

The findings refute the classic perception of innovation presented in the EU in the Oslo Manual of 2005. According to this material, marketing innovations are classified as non-technological innovations, but research has shown that current enterprises perceive marketing innovations as a combination of technological and non-technological innovations. Therefore, marketing innovations in Industry 4.0 cannot be described as non-technological, but they cannot be labeled as merely technological either.

An interesting difference was found in the perception of Industry 4.0 in the secondary research. Some authors, for example Wee et al. [9] and Cooper et al. [8], present Industry 4.0 as an industrial revolution associated with a step-up in productivity gains. The authors showed that new technologies, digitization, and robotization are leading to a revolution in industry. On the other hand, the authors of References [57,81,82] perceive Industry 4.0 as a natural evolution in industry, not revolution. From the presented research we cannot unequivocally favor either evolution or revolution.

If we evaluate the innovative marketing tools associated with Industry 4.0 that have been identified by this research, it is a combination of long-term tools and brand-new tools. Examples of marketing tools that were used before the idea of Industry 4.0 came into being include events, relationship marketing, employer branding, and product placement. Examples of current marketing tools include the Internet of Things, additive production, augmented reality, virtual reality, and virtual currency. When comparing the identified innovative marketing tools from secondary research with the findings from primary research, there was a strong consensus. The identified innovative marketing tools identified from secondary tools coincided with primary research. Research has shown that there is no boundary between Industry 4.0 and innovative marketing. The identified list of seventeen tools was a compilation of Industry 4.0 as perceived by the authors of References [34–37,40,41,43].

The fundamental local effect of any industrial revolution is the rapid increase in labor productivity, as confirmed by References [83–85]. In the presented research, enterprises also identified an increase in labor productivity, but this ranked sixth in the ranking of importance. The most important impacts were "entering new markets, increasing competitiveness", which are connected with growth strategies. The third strongest impact of Industry 4.0 implementation was "building PR and a brand". Many enterprises today present themselves as implementing Industry 4.0, but even the presentation of business involvement in Industry 4.0 is often just a marketing strategy. In this way, the enterprise presents itself is connected with innovation, but the reality is completely different.

The relationship of innovative marketing in Industry 4.0 and the impact that this implies is essential for practical use in strategic planning. Identified impacts that may be strategic goals in strategic planning as stated by Müller et al. [86] and Wolf and Floyd [87] can be fulfilled with a marketing strategy in the form of a precisely designed marketing mix. On the other hand, marketing innovations need to be seen only as part of the business strategy, along with product, process, and organizational innovations.

## 6. Conclusions

The reason for carrying out the research presented in this paper was to find out the current perception of Industry 4.0 in marketing practice. Theoretically, this topic has been written about quite often, as shown by the secondary research, but there is a lack of a practical perspective coming directly from enterprises. This paper provided this enterprise view in which the main research question was "How to use current Industry 4.0 innovative marketing tools for enterprises practice?" The authors conducted a pilot study, the results of which followed the primary research. We managed to identify innovative marketing tools and sort them by importance. The next step was to identify effects and to rank them by their impact on enterprises. Finally, the tools were confronted with impacts and dependent pairs were constructed using correlation analysis. This correlation was used in the preparation of specific marketing mixes for individual strategic objectives. Every enterprise plan and strategic goal are a part of long-term plans. For these goals, marketing mixes identified in the field of marketing, consisting of innovative tools, which according to companies belong to Industry 4.0, can be

used. This conclusion can answer the main research question. Three other research questions were answered in the research process.

The main importance of the presented research is the possibility of applying the results in practice. The results obtained can be used as guidelines for drawing strategic plans. A large number of strategic objectives and the ways to achieve them were identified. Enterprise managers can use the marketing mix discovered by the research, which is likely to be effective, for achieving these goals. Of course, the use of only marketing to fulfill the enterprise strategy would be insufficient. The importance of the paper is also in the theoretical area, where a comprehensive overview of the current Industry 4.0, as perceived at present, was recorded.

This paper is highly extensive and comprehensive, in that it combines several studies and places them into logical contexts. Even so, there are certain limitations to the results it presents due to the scope. This paper does not present the differences resulting from the sorting parameters of the enterprises, such as the size of the enterprise, sector or ownership. This could form the subject of future research. The research was conducted in 2018/19; however, the situation in the sector in question is developing very rapidly, and so the same research method should be applied regularly, such as every two years. The research may serve as a basic record of situations that could be the subject of further research using the same methodology. The paper presents a combination of seventeen innovative marketing tools and eleven effects that they cause. Each of the tools and impacts is described briefly in the paper, but a more precise and detailed explanation could be another possible direction on which researchers could focus. The greatest benefit of the paper is that the results may be used in practice for strategic planning. Verifying the accuracy of the compiled proposals is the biggest challenge for marketing researchers in the era of Industry 4.0.

In conclusion, the research results showed that emerging opportunities from Industry 4.0 are positive drivers for growth strategies and increased business competitiveness.

**Author Contributions:** Conceptualization, O.U. and J.D.; methodology, O.U.; software, O.U.; validation, O.U. and J.D.; formal analysis, J.D.; investigation, J.D.; resources, O.U.; writing—original draft preparation, O.U.; writing—review and editing, O.U.; visualization, J.D.; supervision, O.U.; project administration, J.D.; funding acquisition, J.D.

**Funding:** This research received no external funding.

**Acknowledgments:** This article was written as part of a project to support research teams of excellence at the Faculty of Economics of the Technical University of Liberec, which is financed by institutional support for long-term conceptual development at the Faculty of Economics of the Technical University of Liberec.

**Conflicts of Interest:** The authors declare no conflict of interest.

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
