# Peer review of "Marketing Innovations in Industry 4.0 and Their Impacts on Current Enterprises"

_applsci, doi:10.3390/app9183685_

Round 1

Reviewer 1 Report

Dear authors, thank you for giving me the chance to review your paper. I believe that you tackle an interesting subject, whereas the paper could be further improved according to the following comments:

- The introduction and literature review refers to innovation and marketing regarding Industry 4.0. What is missing here is the link to business model innovation, that is currently missing in the literature review. A lot of authors refer to "customer channels" or terms alike regarding marketing in the context of innovation and Industry 4.0. Please refer to the literature below, among further, in order to extend the literature base and discussion:

Ibarra, D., Ganzarain, J., & Igartua, J. I. (2018). Business model innovation through Industry 4.0: A review. Procedia Manufacturing, 22, 4-10.

Ibarra, D., Igartua, J. I., & Ganzarain, J. (2019). Business model innovation from a technology perspective: a review. In Engineering Digital Transformation (pp. 33-40). Springer, Cham.

Müller, J. M., Buliga, O., & Voigt, K. I. (2018). Fortune favors the prepared: How SMEs approach business model innovations in Industry 4.0. Technological Forecasting and Social Change, 132, 2-17.

Müller, J. M., & Däschle, S. (2018). Business Model Innovation of Industry 4.0 Solution Providers Towards Customer Process Innovation. Processes, 6(12), 260.

Müller, J. M. (2019). Business model innovation in small-and medium-sized enterprises: Strategies for industry 4.0 providers and users. Journal of Manufacturing Technology Management.

Rachinger, M., Rauter, R., Müller, C., Vorraber, W., & Schirgi, E. (2018). Digitalization and its influence on business model innovation. Journal of Manufacturing Technology Management.  

- The sample, how it was selected, and some information (industry sectors, distribution among smaller/larger enterprises etc.) could be given.

- The last section could better reflect on limitations and suggestions for future research.

Some minor remarks:

- Reference No. 76: Rüssmann (spelling mistake)

- Table 5 could be formatted better (text aligned instead of justified), Figure 1: The g of Marketing is cut off.

Author Response

Comments and Suggestions for Authors

Dear authors, thank you for giving me the chance to review your paper. I believe that you tackle an interesting subject, whereas the paper could be further improved according to the following comments:

- The introduction and literature review refers to innovation and marketing regarding Industry 4.0. What is missing here is the link to business model innovation, that is currently missing in the literature review. A lot of authors refer to "customer channels" or terms alike regarding marketing in the context of innovation and Industry 4.0. Please refer to the literature below, among further, in order to extend the literature base and discussion:

Added. We used the recommended literature.

Ibarra, D., Ganzarain, J., & Igartua, J. I. (2018). Business model innovation through Industry 4.0: A review. Procedia Manufacturing22, 4-10.

Ibarra, D., Igartua, J. I., & Ganzarain, J. (2019). Business model innovation from a technology perspective: a review. In Engineering Digital Transformation (pp. 33-40). Springer, Cham.

Müller, J. M., Buliga, O., & Voigt, K. I. (2018). Fortune favors the prepared: How SMEs approach business model innovations in Industry 4.0. Technological Forecasting and Social Change132, 2-17.

Müller, J. M., & Däschle, S. (2018). Business Model Innovation of Industry 4.0 Solution Providers Towards Customer Process Innovation. Processes6(12), 260.

Müller, J. M. (2019). Business model innovation in small-and medium-sized enterprises: Strategies for industry 4.0 providers and users. Journal of Manufacturing Technology Management.

Rachinger, M., Rauter, R., Müller, C., Vorraber, W., & Schirgi, E. (2018). Digitalization and its influence on business model innovation. Journal of Manufacturing Technology Management.  

- The sample, how it was selected, and some information (industry sectors, distribution among smaller/larger enterprises etc.) could be given.

Added. The selection of the sample described in detail.

- The last section could better reflect on limitations and suggestions for future research.

Added.

 Some minor remarks:

- Reference No. 76: Rüssmann (spelling mistake)

- Table 5 could be formatted better (text aligned instead of justified), Figure 1: The g of Marketing is cut off.

corrected

Reviewer 2 Report

My comments are following:

in line 263 should be , instead of " in line 274 and 275 there is unnecessary space lines 297-307, it would be good to know the sample charcteristics, was it representative sample? what was the population and sample selection? line 304, SPSS program or package instead of SPSS statistics line 312 quantitative variables instead quantitative quantities figure 1: 2. research question - spelling, font size line 383, guerilla- spelling line 437, more information about qualitative research from pilot project would be nice, reader will not have to look for your previous paper line 505-520, Wilcoxon test is not a right tool to answer this question. I recommend to skip this part. Results from this test are not sufficient to the conclusions you give. lines 539-543, was the sample randomly choosen? if the sample is not random sample it is not the right way to assess the correlation based on the p-value. you can only assess the r value and measure the strenght and direction of the correlation, not the significance line 605: " line 616: pan-European ? paragraph 616-621 should be described at the begining of the paper when you introduce the survey and the sample I did not checked the References, are all 81 positions cited in the text?

Author Response

My comments are following:

in line 263 should be , instead of "

corrected

in line 274 and 275 there is unnecessary space

corrected

lines 297-307, it would be good to know the sample charcteristics, was it representative sample? what was the population and sample selection?

corrected

line 304, SPSS program or package instead of SPSS statistics

corrected

line 312 quantitative variables instead quantitative quantities

corrected

figure 1: 2. research question - spelling, font size

corrected

line 383, guerilla- spelling

corrected

line 437, more information about qualitative research from pilot project would be nice, reader will not have to look for your previous paper line 505-520,

corrected

Wilcoxon test is not a right tool to answer this question. I recommend to skip this part. Results from this test are not sufficient to the conclusions you give.

Removed from the text.

 lines 539-543, was the sample randomly choosen? if the sample is not random sample it is not the right way to assess the correlation based on the p-value. you can only assess the r value and measure the strenght and direction of the correlation, not the significance

The respondents were selected using systematic random sampling, which has to be performed in order to enable statistical induction tests to be used. The parent university has access to the Bisnode database of companies, where enterprises are sorted alphabetically only, and which contains the entire population. The system used to select the enterprises involved contacting every 50th enterprise, which complied with the conditions of random research and enabled further statistical processing.

line 605: " line 616: pan-European ?

corrected

 paragraph 616-621 should be described at the begining of the paper when you introduce the survey and the sample I did not checked the References, are all 81 positions cited in the text? 

We moved the paragraph to the Introduction.

All the references were checked and added according to the sample.

Round 2

Reviewer 1 Report

Thank you for your revisions, I believe that the paper is publishable in the current version.

Reviewer 2 Report

No comments.